behaviour/ecology/environmental science

bowhead whale, migration, acoustic monitoring, sea ice

**Author for correspondence:**
S. J. Insley
e-mail: sinsley@wcs.org

# Bowhead whales overwinter in the Amundsen Gulf and Eastern Beaufort Sea

S. J. Insley[1,2], W. D. Halliday[1,3], X. Mouy[3,4] and N. Diogou[1,3]

[1]Wildlife Conservation Society Canada, Whitehorse, Yukon, Canada
[2]Department of Biology, and [3]School of Earth and Ocean Sciences, University of Victoria, Victoria, British Columbia, Canada
[4]JASCO Applied Sciences Ltd, Victoria, British Colombia, Canada

SJI, 0000-0003-3402-8418; WDH, 0000-0001-7135-076X;
XM, 0000-0003-4938-1214; ND, 0000-0001-9160-7120

The bowhead whale is the only baleen whale endemic to the Arctic and is well adapted to this environment. Bowheads live near the polar ice edge for much of the year and although sea ice dynamics are not the only driver of their annual migratory movements, it likely plays a key role. Given the intrinsic variability of open water and ice, one might expect bowhead migratory plasticity to be high and linked to this proximate environmental factor. Here, through a network of underwater passive acoustic recorders, we document the first known occurrence of bowheads overwintering in what is normally their summer foraging grounds in the Amundsen Gulf and eastern Beaufort Sea. The underlying question is whether this is the leading edge of a phenological shift in a species' migratory behaviour in an environment undergoing dramatic shifts due to climate change.

## 1. Introduction

Climate change is predicted to alter animal migratory movements, primarily timing and range extensions [1,2], although the degree of phenotypic plasticity appears to be taxonomically specific [3–5]. Some seasonally migrating species are known to vary the extent and timing of their movements in relation to current environmental conditions [6,7] which may be related to differences in environmental variability as well as what is driving the migratory behaviour in the first place [5,8]. Of particular importance to the conservation of many species in the current context of climate change is the degree to which phenotypic plasticity related to migratory behaviour facilitates adaptation to a warming world [9].

Cetaceans, particularly baleen whales, are well known long-distance migrants, many species having some of the longest

**Figure 1.** Map of the Amundsen Gulf and eastern Beaufort Sea area showing the locations of the four passive acoustic recorders, which provided data for the reported observations during the 2018–2019 winter as well as the locations of earlier recording sites (Sachs Harbour and Ulukhaktok) used for comparisons.

known regular migrations of any taxa [10] but with occasional anomalies and regular variations in the destination and distance of migratory patterns known to exist [11–14]. The bowhead whale (*Balaena mysticetus*) is the only Arctic endemic baleen whale and is well adapted physically to suit this environment [15–17]. The species is exceptionally long lived (maximum greater than 200 years), slow to reproduce (3–4 year birthing interval with 13–14 month gestation) and primarily forages on macro zooplankton in the water column as well as near-bottom on epibenthic organisms [17–20]. Historically, there is evidence that bowhead whale movements spanned the Canadian Arctic archipelago [21]. Currently, the four recognized management stocks of bowhead whales circum-Arctic are in the (i) Okhotsk Sea, (ii) Bering-Chukchi–Beaufort Seas, (iii) Eastern Canada–West Greenland and (iv) Svalbard/Barents Sea [22]. While there is at least occasional overlap between stocks [23], the extent of mixing is unknown. The subject of our investigation is the Bering–Chukchi–Beaufort (BCB) stock, which is the largest (*ca* 20 000) and along with the Eastern Canada–West Greenland stock, the only populations that are stable or growing in number [24,25]. Seasonal movement patterns of the BCB bowhead population are relatively well understood through several years of detailed satellite telemetry studies and although there is substantial within and among individual variation, like many migratory species, there is a clear annual migratory pattern [26–28]. BCB bowheads typically overwinter in the north Bering Sea, begin to migrate northward and then eastward in early spring as ice leads (i.e. channels of open water) form, and spend the summer foraging in the eastern Beaufort Sea including the Amundsen Gulf and north of Banks Island in the Viscount Melville Sound (figure 1). The whales return to the Bering Sea in the fall, completing an approximate 6000 km annual migration [26–30].

As far as we know, prior to the winter of 2018–2019 there is no record or indication, scientific or Indigenous Knowledge (Richard J. Kudlak, Joseph Kitekudlak, personal communication, July 2019, Ulukhaktok NT), that bowheads had previously overwintered in the eastern Beaufort Sea. The 2018–2019 winter was characterized by large areas of open water (i.e. ice-free), in the east Amundsen Gulf. During this period, several sightings of bowheads were made throughout the winter by residents near the community of Ulukhaktok in the eastern extent of the Amundsen Gulf (Jared and Jordan Kitekudlak, personal communication, July 2019, Ulukhaktok NT). Here, we present evidence from

several passive acoustic underwater recordings which confirm and provide more detail on these sightings, providing evidence of bowhead whales overwintering in the Beaufort Sea and Amundsen Gulf, a migratory anomaly that could be directly linked to climate change.

# 2. Methods

## 2.1. Acoustic recordings

We collected passive acoustic data at four locations in the south Amundsen Gulf during the 2018–2019 winter (figure 1) and compared these to several of our previous recordings in the region (see *Comparisons with Other Datasets* below). We deployed a Wildlife Acoustics SM3M acoustic recorder (Maynard, Massachusetts, USA) fitted with a low-noise hydrophone (High Tech, Inc., Gulfport, Mississippi, USA) between July 2018 and July 2019 near Ulukhaktok (70°42.857′ N, 117°48.020′ W). The acoustic recorder was set to a 48 kHz sample rate with 16-bit depth, a duty cycle of 5 min of recording every hour, and +12 dB of gain. The positively buoyant acoustic recorder was suspended roughly 3 m above the sea floor in 30 m of water and stopped recording on 24 April 2019 after running out of power. We also deployed three ST500 acoustic recorders (Ocean Instruments NZ, Auckland, New Zealand) between September 2018 and September 2019 in the south Amundsen Gulf near Pearce Point (70°12.055′ N, 123°09.470′ W) in 350 m of water, and two sites near Cape Bathurst, one in 50 m of water (70°34.5469′ N, 127°39.6319′ W) and one in 300 m of water (70°40.8732′ N, 126°52.2977′ W). These recorders were suspended approximately 10 m above the sea floor by a vertical mooring line with subsurface floats and also recorded one 5-min file every hour at a 48 kHz sample rate.

## 2.2. Bowhead whale acoustic detections

We processed all acoustic data using an automated detector and classifier for Arctic marine mammals (software: Spectro Detector; JASCO Applied Sciences Ltd, Victoria, BC, Canada), which we have used to successfully detect and classify bowhead whale vocalizations, and also beluga whales and bearded seals, in the western Canadian Arctic on multiple datasets [31–34]. The detector is fully described in Mouy *et al.* [35]. Briefly, the software detects acoustic signals between 10 Hz and 8 kHz, and extracts salient features for each signal, such as bandwidth and duration, calculated as in Fristrup & Watkins [36] and Mellinger & Bradbury [37]. A random forest classifier [38] is then used to assign each acoustic signal to a species. The classifier was trained on manually annotated passive acoustic data from the Chukchi Sea [35,39]. 'Manual' analysis refers to the process of aural and visual signal verification and categorization by a trained analyst, as compared to an automated process using classification algorithms.

We performed manual bioacoustic analyses using Raven Pro Software (v. 1.5 and 1.6; Bioacoustics Research Program 2017) by manually inspecting spectrograms (window size: 7000 samples, DFT size: 8192 samples, overlap: 50%) with the time axis set to 15 s and the frequency axis from 0 to 3000 Hz, although the analyst could zoom in or out in either time or frequency as necessary. We manually verified all automated detections of bowhead whale vocalizations between 1 October 2018 and 15 April 2019 (24 April 2019 for the Ulukhaktok site), and manually analysed approximately 10% of files without bowhead detections (systematically selected as the 10th file in a row without a detection). Bowhead whale moans are identified based on their low-frequency characteristics (typically 50–300 Hz frequency range) and length (typically 0.5–2 s long, but can be up to 5 s long) [40,41]. No other species present in this region make vocalizations similar to bowhead whales [40,41], although occasionally sea ice can make low frequency, short tonal sounds that might confuse the automated detector and can sound similar to bowhead calls (W. Halliday, personal observation, 2020). For all recordings in January through April, we also assessed the file directly before and after any file with automated detections in order to increase our chances of finding all bowhead whale vocalizations recorded over the winter when bowheads are not expected to be in the region. In total, we manually analysed 820 of the 4929 (16.6%) files between 1 October 2018 and 24 April 2019 for the Ulukhaktok data. For Pearce Point, we analysed 486 of the 4682 (10.4%) files between 2 October (day the recorder was deployed) and 15 April. For Cape Bathurst at 50 m depth, we analysed 963 of the 4509 (21.4%) files between 10 October (day the recorder was deployed) and 15 April, and for Cape Bathurst at 300 m depth, we analysed 493 of the 4728 (10.4%) files between 1 October and 15 April. The automated detector counted potential bowhead detections in 191 (3.9%) files at Ulukhaktok, 16 (0.3%) files at Pearce Point, 563 (12.5%) files at Cape Bathurst at 50 m depth, and 29 (0.6%) files at Cape Bathurst at 300 m.

## 2.3. Comparisons with other datasets

We compared trends in winter bowhead whale detections from multiple sites where we had previously collected data from the Amundsen Gulf. More specifically, we examined the final date in the autumn and the first date in the spring when a bowhead vocalization was detected. We previously collected and published results from three datasets: Sachs Harbour (71°55.621′ N, 125°23.447′ W) in 2015–2016 [30] and Ulukhaktok (70°42.857′ N, 117°48.020′ W) in 2016–2017 [31] and 2017–2018 [32]. All three of these acoustic datasets used the same type of acoustic recorder (SM3M) and the same mooring configuration and identical settings as the Ulukhaktok recorder described above, but with a recording duty cycle of one 5-min file recorded every 30 min and +18 dB of gain for the Sachs Harbour recorder.

We used the same automated detector and classifier on all of these datasets, as described above, and manually examined 100% of files with an automated detection for bowhead whales and a minimum of 10% of files without any automated detections in the autumn, winter and spring to determine when the final bowhead vocalization occurred during the autumn and first vocalization in the winter or spring occurred.

## 2.4. Sea ice concentration

We examined trends in winter sea ice concentration from 2013 to 2019 within a 100 km radius around the 2018–2019 acoustic recorder locations near Ulukhaktok, Pearce Point and the Cape Bathurst 50 m site using remotely sensed daily sea ice concentration data in 6.25 × 6.25 km grid cells from the AMSR-2 satellite array [42]. We examined the data between 15 November and 15 April of each winter in order to capture the majority of the time between ice formation and ice break-up every year. We calculated the average daily ice concentration across all grid cells within the 100 km radius around each site in each day. We then compared these values between years in order to quantify localized ice concentration.

# 3. Results

## 3.1. Bowhead whale acoustic detections

We detected bowhead whales in 46 of the 4929 files (1.0%) recorded near Ulukhaktok between 1 October 2018 and 15 April 2019 (figure 2a–d and 3). Of these, 12 were between October and December and 34 were between 1 January and 15 April. Bowhead whales were also detected in 24 of the 4682 files (0.5%) near Pearce Point (12 in October to December, 12 in January to April), in 161 of the 4509 files (3.1%) near Cape Bathurst at 50 m depth (five in October to December, 156 in January to April) and in 47 of the 4728 files (1.0%) near Cape Bathurst at 300 m depth (four in October to December, 43 in January to April) (figure 2). The majority of vocalizations recorded were single moans (figure 3a–c), whereas singing occurred starting in early April at both Cape Bathurst sites, as well as from 19 to 24 April near Ulukhaktok. When the singing started occurring, the majority of 5-min files were full of bowhead calls in what appears to be the first example of bowhead singing that we have recorded in this region (figure 3d).

## 3.2. Comparison with other datasets

Unlike the above dataset, our analysis of other acoustic datasets shows evidence that bowhead whales migrated out of the Amundsen Gulf in previous years. At Sachs Harbour in 2015–2016 and Ulukhaktok in 2016–2017 and 2017–2018, there was a gap of at least three months between the last detection in the autumn and the first detection in the spring. At Sachs Harbour in 2015–2016 the last autumn detection occurred on 30 October and the first spring detection occurred on 29 April (181 days). At Ulukhaktok in 2016–2017, the last autumn detection occurred on 24 December and the first spring detection occurred on 2 April (98 days). At Ulukhaktok in 2017–2018, the last autumn detection occurred on 6 December. Unfortunately, this recorder did not consistently record data after 20 January 2018, and shut off completely on 23 March 2018; however, no bowhead detections were found in the data recorded after 6 December. We, therefore, cannot confirm when the first bowhead vocalization occurred near Ulukhaktok in spring 2018.

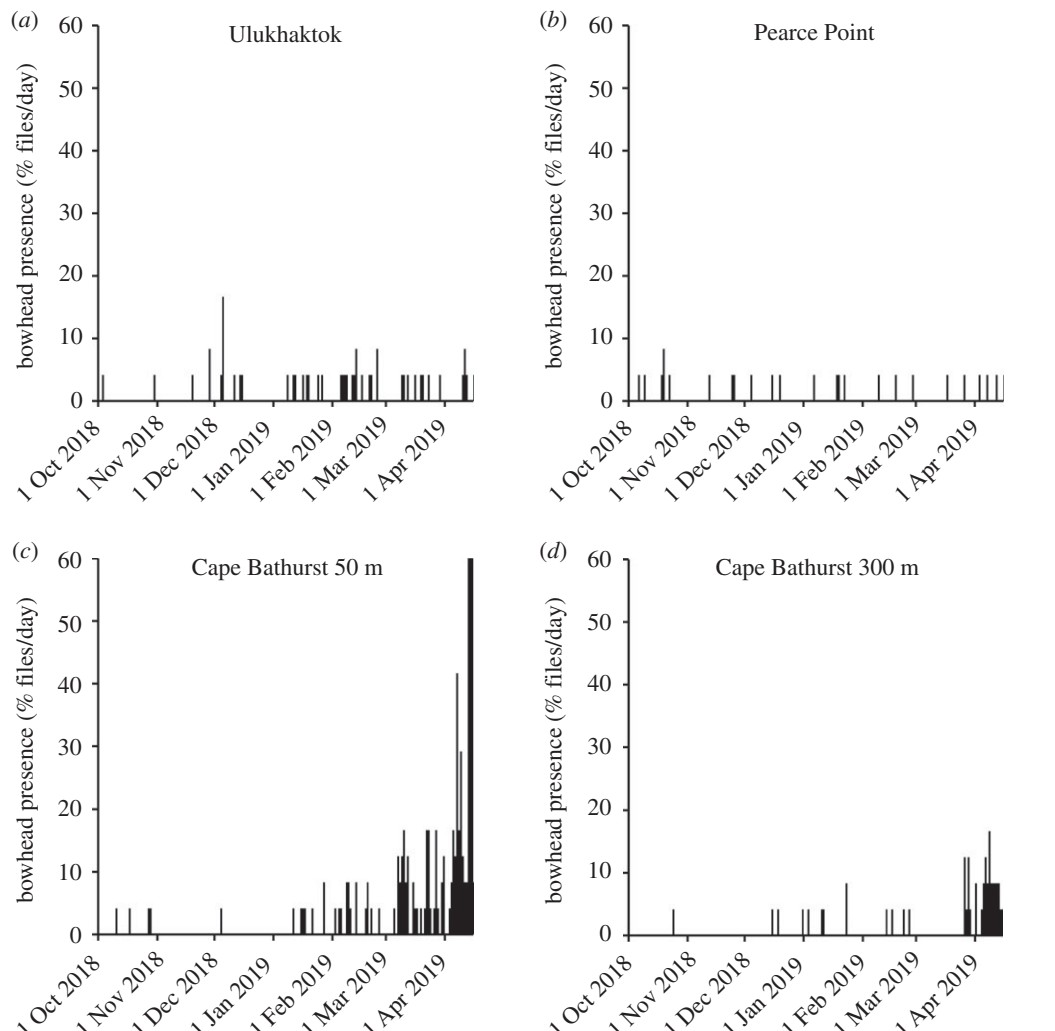

**Figure 2.** (*a*) The presence of bowhead whale vocalizations in acoustic data recorded between October 2018 and April 2019 near (*a*) Ulukhaktok, (*b*) Pearce Point and Cape Bathurst at (*c*) 50 m depth and (*d*) 300 m depth. Bowhead presence is measured as the number of 5-min files per day with bowhead vocalizations divided by the total number of files recorded per day (i.e. [24]).

## 3.3. Sea ice concentration

Average sea ice within a 100 km radius around Ulukhaktok was, on average, lower in 2018–2019 compared to all other years between 2013 and 2018 (Kruskal Wallis $\chi^2 = 45.80$, d.f. = 5, $p < 0.0001$; figure 4). More specifically, the mean for 2018–2019 was 95.4%, but was greater than 97% in all other years. Although 2015–2016 holds the record for the minimum mean ice concentration of 70.4% at Ulukhaktok, 2018–2019 is a close second for this record (72.4%), and other more important markers of the distribution, including median, 1st quartile and 5th percentile, are all lower in 2018–2019 than in all other years. Sea ice within a 100 km radius of Pearce Point was similarly lowest on average in 2018–2019 (Kruskal Wallis $\chi^2 = 63.20$, d.f. = 5, $p < 0.0001$; mean = 93.9%; figure 4) compared with the previous five winters (greater than or equal to 95.9%). Sea ice within a 100 km radius of Cape Bathurst was significantly lower in 2018–2019 (Kruskal Wallis $\chi^2 = 95.55$, d.f. = 5, $p < 0.0001$; mean = 90.7%; figure 4) compared with 2013–2014 (96.4%), 2014–2015 (94.4%) and 2016–2017 (95.4%), but was not significantly different than either 2015–2016 (92.2%) or 2017–2018 (92.7%).

## 4. Discussion

The evidence is clear that BCB bowheads overwintered in their summer foraging region in the eastern Beaufort Sea and Amundsen Gulf during the 2018–2019 winter and as far as we know, this is the first time it has been reported. Bowhead whales remained in the Ulukhaktok and Pearce Point areas for

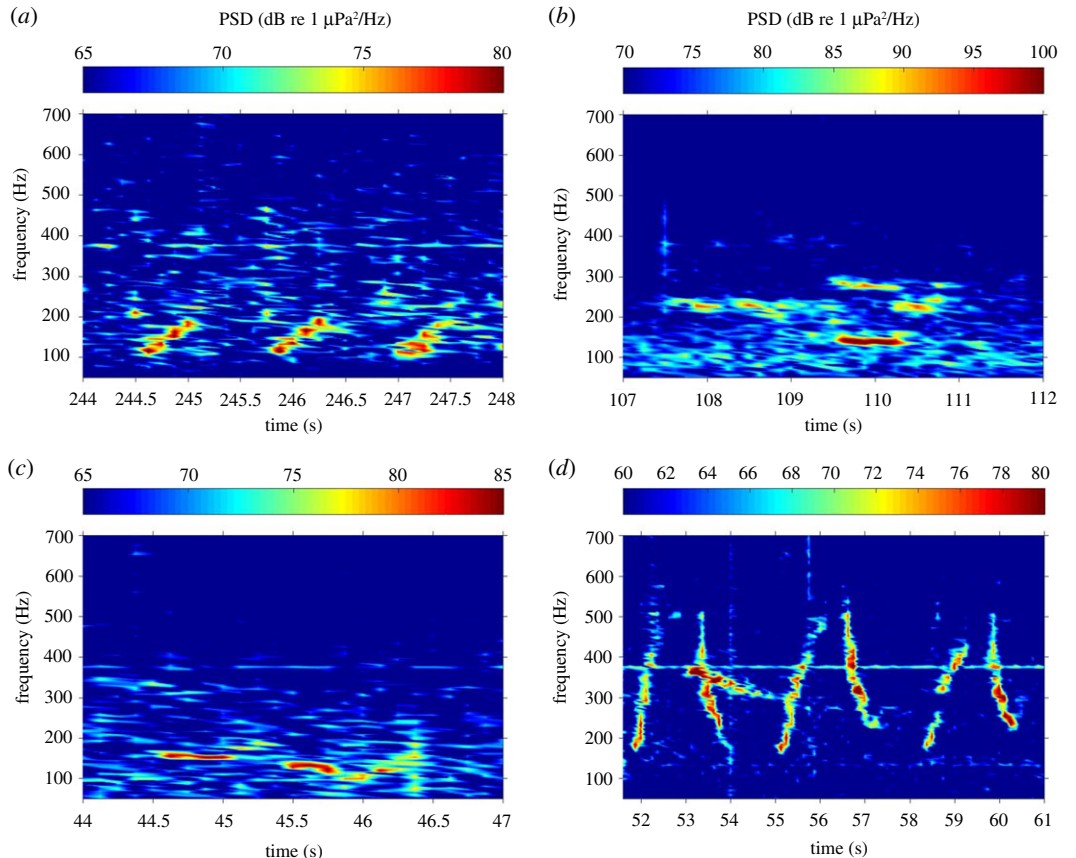

**Figure 3.** Examples of winter vocalizations of bowhead whales at Ulukhaktok, Northwest Territories, Canada recorded during the winter of 2018–2019. (a) Upsweep moans recorded 4 December 2018; (b) moans recorded 8 February 2019; (c) moans recorded 11 February 2019 and (d) multiple combination upsweep/downsweep moans recorded 19 April 2019, possibly an example of singing (the two-unit sequence was repeated continuously for greater than 5 min). Spectrograms are measured using power spectral densities (PSD); note that each panel has different scales for PSD.

the entire winter and part of the winter off Cape Bathurst. Our understanding is that there are no Indigenous Knowledge records of such an event happening in the past, supporting this being a first-time occurrence (Richard J. Kudlak, Joseph Kitekudlak, personal communication, July 2019, Ulukhaktok NT). The bowhead detections from different sites overlap very closely in time and the distances between most sites are likely too far for the propagation of bowhead vocalizations (208 km between the Ulukhaktok and Pearce Point recorders, 149 and 173 km between the Pearce Point and Cape Bathurst recorders, and 334 and 364 km between the Ulukhaktok and Cape Bathurst recorders; but, 32 km between the two Cape Bathurst recorders), making it likely that the calls detected at different sites were from different whales and that this was a broader phenomenon than a small number of whales near Ulukhaktok that were sighted locally (Jared and Jordan Kitekudlak, personal communication, July 2019, Ulukhaktok NT). For example, bowhead vocalizations were detected at both Cape Bathurst and Ulukhaktok within 7 h of each other on 11–12 January (11 January at 20.00 UTC at Cape Bathurst at 300 m depth and 12 January at 3.00 UTC at Ulukhaktok; distance = 334 km), at both Cape Bathurst and Pearce Point within 4 h of each other on 17–18 January (17 January at 22.00 UTC at Cape Bathurst at 50 m and 18 January at 2.00 UTC at Pearce Point; distance = 173 km), and again at both Cape Bathurst and Ulukhaktok within 1 h of each other on 8 February (19.00 UTC at Ulukhaktok and 20.00 UTC at Cape Bathurst at 300 m depth; distance = 334 km). The exceptionally low background noise during winter does make it possible that the calls could be detected at both of our closest sites (i.e. the two Cape Bathurst sites) [32,33]. Cape Bathurst would seem to be the more likely occasional overwintering location, being both a known summertime foraging congregation location for bowhead whales, particularly sub-adults, and a wintertime polynya [28]. Our recordings indicate that bowheads left the Cape Bathurst area in the fall, with a minimum of one month between October and November with no detections, but were detected there again every month of the winter

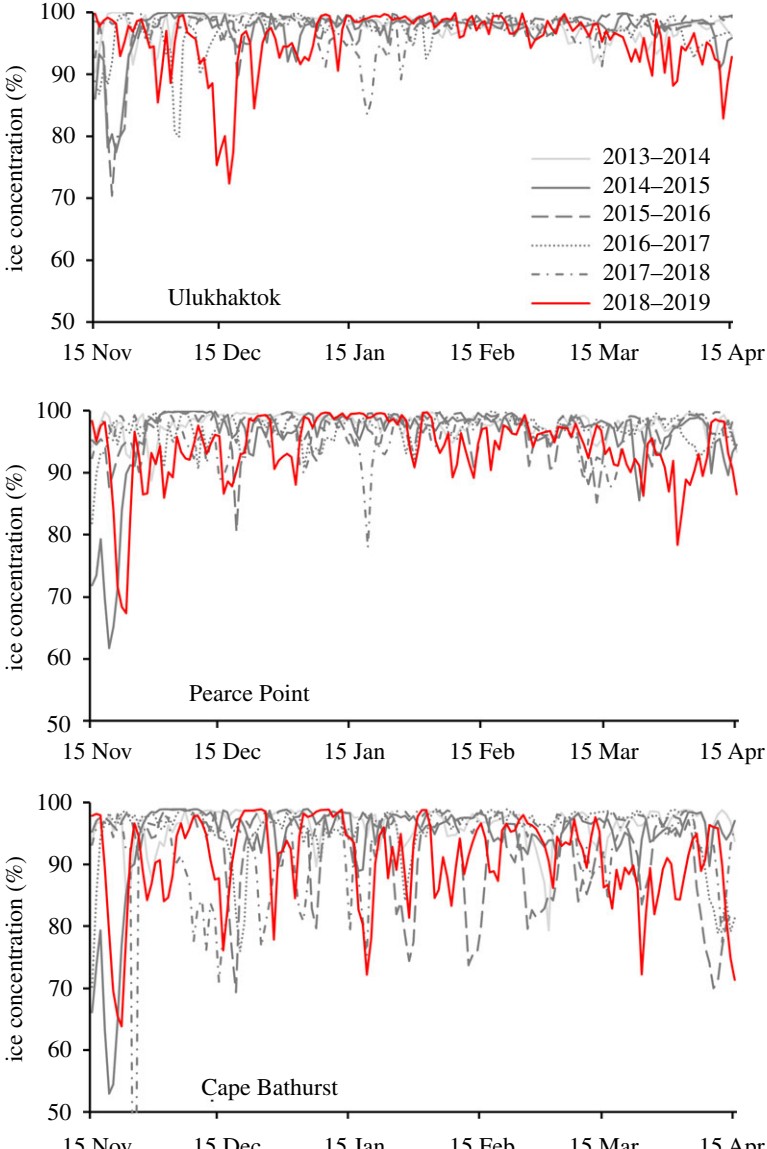

**Figure 4.** Mean daily sea ice concentration within 100 km of the acoustic recorder near Ulukhaktok, Pearce Point and Cape Bathurst (50 m site), Northwest Territories, Canada, between 15 November and 15 April of each winter from 2013 to 2019.

between December and April. Whether or not the winter bowhead whales at Bathurst or Pearce Point were the same individuals as those who left Bathurst in October is unknown.

There are several potential proximate drivers for the whales overwintering in the Amundsen Gulf. Ice entrapment has been documented with whales that were not ice-adapted [43], but is unlikely here as bowhead whales are known to move about in heavy ice conditions [16,30] and the east Amundsen Gulf had more ice-free water than normal during the winter of 2018–2019 (figure 4). Parasite and disease avoidance and molt have been raised as explanations driving whale migration [44,45] but seem less applicable for BCB bowheads whose migration is long (*ca* 6000 km) but largely east-west with relatively constant near-zero water temperatures. Predator avoidance is also worth considering. With warming ocean temperatures leading to decreased sea ice cover, killer whales (*Orcinus orca*) have been observed more regularly in the Chukchi and Alaskan Beaufort Seas [46] and are known to influence bowhead movement in the eastern Canadian Arctic [47]. Given the low levels of killer whale activity in the eastern Beaufort relative to the Bering and Chukchi Seas, killer whale presence could have been among the factors that led bowheads to remain in the eastern Beaufort Sea and Amundsen Gulf over winter in 2018–2019.

The potential of energetic savings from not migrating would need to be balanced with any change in food intake in order to result in a net benefit. Foraging that may have occurred during the fall migration

or during the winter in the Bering Sea would need to be compensated with opportunities in the eastern Beaufort Sea, less the energetic cost of migrating. Bowhead preferred prey, such as copepods (*Calanus spp.*), are known to aggregate while hibernating near the sea floor and may have been available during winter if not too deep [19]. Overwintering would also ensure bowheads were present for the early summer plankton bloom, which may be limited in duration and occurring earlier and less predictably as the increasingly ice-free water allows greater light penetration into the water column [24]. In addition, if the overwintering whales were sub-adults of pre-reproductive age, as has been shown in other cetacean species [48], they would likely be smaller in size and thus may require less food intake than breeding adults.

The relationship between water temperature and thermoregulation, independent of the water temperature–prey density relationship [7], could also be driving overwintering. Chambault *et al.* [49] found evidence for a narrow temperature range preference window between −0.5 and 2°C with the Eastern Canada—West Greenland stock of bowheads that was apparently more important than foraging and driven by the risk of hyperthermia when active (e.g. migrating) in warm water. Citta *et al.* [27] documented what may have been such an avoidance of the warm coastal waters associated with the Alaska Coastal Current by BCB bowheads. Anomalously high ocean temperatures also appear to be occurring with increased frequency. The summer following the overwintering observations described here, a widespread aerial census of bowhead whales in the Beaufort Sea conducted between July and October 2019 indicated unusually late westward migrating whales, many found further offshore than is normally the case [50]. It is not clear if this pattern of movement was attributable to warmer water temperatures or related to the 2018–2019 overwintering event. Such strong avoidance of warm waters have also been recently demonstrated in narwhals in the eastern Arctic [51,52]. If the avoidance of warm ocean temperatures were the primary driver of this anomalous behaviour, it may be a significant warning sign for bowhead whales. With this in mind, along with maintaining and expanding the locations of long-term recordings, our next step will be to correlate these data with ocean temperatures and other important oceanographic variables in the region.

Ethics. All data were collected under the authority of the Aurora Research Institute Scientific Research Licence No. 15996.

Data accessibility. Dryad data link: https://doi.org/10.5061/dryad.rn8pk0p7n.

Authors' contributions. S.J.I. and W.D.H. conceived of and designed the study; S.J.I. and W.D.H. collected all field data; S.J.I., W.D.H., X.M. and N.D. conducted all data analysis; S.J.I., W.D.H., X.M. and N.D. wrote the manuscript; S.J.I., W.D.H., X.M. and N.D. have read and approved the manuscript before submission; S.J.I., W.D.H., X.M. and N.D. gave final approval for publication and agree to be held accountable for the work.

Competing interests. We declare we have no competing interests.

Acknowledgements. We are grateful to the people of Ulukhaktok for working with us, especially A. Kudlak, B. Inuktalik and the Olokhaktomiut Hunters and Trappers Committee. Thanks also to the captain and crew of the CCGS Sir Wilfrid Laurier and especially Humfrey Melling and Andrea Niemi for assistance with our at-sea moorings and for the detailed and thoughtful comments provided by two anonymous reviewers. Funding was provided by The W. Garfield Weston Foundation, the Fisheries Joint Management Committee of the Inuvialuit Settlement Region, a Mitacs Elevate Postdoctoral Fellowship to N.G. and Fisheries and Oceans Canada. All data were collected under the authority of the Aurora Research Institute Scientific Research Licence No. 15996.

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
