## [Peer Review File · Royal Society Open Science]

Review History

RSOS-202268.R0 (Original submission)

Review form: Reviewer 1

Is the manuscript scientifically sound in its present form?

Yes

Are the interpretations and conclusions justified by the results?

Yes

Is the language acceptable?

Yes

Do you have any ethical concerns with this paper?

No

Have you any concerns about statistical analyses in this paper?

No

Recommendation?

Accept with minor revision (please list in comments)

Comments to the Author(s)

Bowhead Whales Overwinter in the Amundsen Gulf, Eastern Beaufort Sea

Review. January 13, 2021

I was a reviewer of the original submission to Biology Letters, and have carefully reviewed this next version and the authors responses to reviewer comments. In general, I think they have done a good job addressing all of the various comments, and I found that the toning down of the speculation improved the paper.

As written, title implies that Amundsen Gulf is part of the Eastern Beaufort Sea, and it is my understanding they are separate, adjacent water bodies.

Ice

There is still one part of this revised paper that is troubling me, that is the sea ice analysis. The analysis in this version is different than the previous; in this round, they make quantitative comparisons based on a 6 x 6 km grid cell scale, do not appear to define the boundaries of the study area (I may have missed that), and as before, they still base the analysis on only five years of sea ice data. Using these time and space scales is not well relevant to our understanding of bowhead winter ecology and movements, or, to variation in sea ice that is well documented for this region (takes decades to elucidate statistical change/shifts).

Shore to shore to shore fast ice throughout Amundsen Gulf during winter, which is the most common winter condition, could be expected to limit or deter bowheads from entering and overwintering into east Amundsen Gulf. This is inferred from published observations of tagged bowheads 'waiting' at 10/10 ice edges in the Bering Sea and off Cape Bathurst in spring, yet easily navigating cracks etc. in sea ice of 9+ . Therefore, the 6 km x 6 km grid cell scale is not relevant or helpful considering a bowhead whale in winter habitat, while the entirety of the Gulf and ice conditions therein is likely more relevant.

It would be better to say that when there is limited or no fast ice covering the Gulf (e.g., as was the case for the winter of 2018-19 except for 7 days of continuous ice cover (and incidentally in 2017/2018 also), bowheads would have access to eastern Amundsen Gulf in winter. So while winter 2018/2019 sea ice conditions do not appear to have included an extensive or obvious barrier to bowhead overwintering near Ulu, it is an important point that neither did the year before, and neither did about 10 winters over the last four decades.

Specific comments (page numbers relate to the marked up version, not the clean version)

Line 60: Change "Traditional" to "Indigenous"

Line 67 change "to" to "which"

Line 69 change "likely to be" to "could be", or, end sentence at anomaly.

Lines 101, 105, 107, 118 several other instances - is the term manually defined? I'm not exactly sure I know what you mean here. Visual inspection of a computer screen? Measured?

Lines 118-124 : all these additions - could this go in a table? Comes across as a list of data

Line 142, 184: must indicate winters as 2013/2014 etc. etc. don't give just one year for a winter - several instances

Line 147: "always" and "right before break up" - what is the scale of the study area you are referring to? can't possibly be in every 6 km x 6 km grid cell? At that fine scale, how can you say if break up had occurred at a scale relevant to bowheads.

What is the rationale for a grid cell of that size and how is that relevant to bowhead whales.

Line 206: insert number of days not just date

Line 218: ...possible that the same calls could be detected at both of our closest sites.....

Line 219: revise as follows (part struck out is irrelevant here)

.....Cape Bathurst would seem to be the more likely occasional overwintering location, being both a known summertime foraging congregation location for bowhead whales, particularly sub-adults, and a wintertime polynya (27).

Line 233: insert "Gulf" after Amundsen.

Review form: Reviewer 2

Is the manuscript scientifically sound in its present form?

Yes

Are the interpretations and conclusions justified by the results?

Yes

Is the language acceptable?

Yes

Do you have any ethical concerns with this paper?

No

Have you any concerns about statistical analyses in this paper?

No

Recommendation?

Accept with minor revision (please list in comments)

Comments to the Author(s)

This paper is much improved compared to the first version. I only have minor comments, pasted below:

Line 16, abstract: I suggest deleting "not completing their annual migration" and sticking with what you have high certainty about, i.e., the fact that bowheads overwintered in what is normally their summering grounds.

Lines 27-28: "which may be related to how dynamic and the magnitude of ..." please improve phrasing

Line 37: "(maximum >200 years)" Such an extreme statement requires a citation.

Line 50: "... and then eastward in early spring as ice leads form," ("in" is missing)

Lines 54-56: somewhat clumsy sentence, suggest at least a comma after "of": "... that we are aware of, that bowheads had ..." Even better would be: "As far as we know, prior to the winter of 2018/19 there is no record or indication, scientific or Traditional Knowledge (...), that bowheads had previously overwintered in the Beaufort Sea."

Line 68: "four locations in the South Amundsen Gulf" See comment below regarding Figure 1.

Line 69: remove "s" on recorders.

Lines 109-113: would be nice to have the percentages of files analyzed for each recorder, much easier to assess in a glance than these numbers.

Lines 114-116: same here, what percentage of the files analyzed at Ulukhaktok are 191 files, etc ... ?

Line 210: isn't "the Amundsen" colloquial?

Line 256: it's always nice when authors acknowledge the reviewers' work

Figure 1: since the place names on the map are duplicated in the legend, which is unnecessary, it seems to me it would be much more helpful to the reader to use one symbol for the 2018-19 recorders and another for the historical recorders.

Decision letter (RSOS-202268.R0)

Dear Dr Insley

On behalf of the Editors, we are pleased to inform you that your Manuscript RSOS-202268 "Bowhead Whales Overwinter in the Amundsen Gulf, Eastern Beaufort Sea" has been accepted for publication in Royal Society Open Science subject to minor revision in accordance with the referees' reports. Please find the referees' comments along with any feedback from the Editors below my signature.

Please submit your revised manuscript and required files (see below) no later than 7 days from today's (ie 15-Mar-2021) date. Note: the ScholarOne system will 'lock' if submission of the revision is attempted 7 or more days after the deadline. If you do not think you will be able to meet this deadline please contact the editorial office immediately.

on behalf of Dr Denise Greig (Associate Editor) and Kevin Padian (Subject Editor)
openscience@royalsociety.org

Associate Editor Comments to Author (Dr Denise Greig):

Associate Editor: 1

Comments to the Author:

I enjoyed reading this interesting paper and have a few minor questions. I defer to the reviewers experience and knowledge of bowhead whales for a more detailed review and think this will be an interesting addition to the literature once their concerns have been addressed.

I was curious why acoustic recorders were deployed in this region – was bowhead presence suspected?

In the discussion on page 10, line 200. Where it says a small number of whales were sighted locally, can you please elaborate here? When were Ulukhahtok whales sighted? Are these sightings backing up your results (i.e. sightings of whales overwintering in their summer feeding area in addition to the sounds)?

And, finally, I am curious what the significance is of the possible bowhead songs that you recorded.

Reviewer comments to Author:

Reviewer: 1

Comments to the Author(s)

Bowhead Whales Overwinter in the Amundsen Gulf, Eastern Beaufort Sea

Review. January 13, 2021

I was a reviewer of the original submission to Biology Letters, and have carefully reviewed this next version and the authors responses to reviewer comments. In general, I think they have done a good job addressing all of the various comments, and I found that the toning down of the speculation improved the paper.

As written, title implies that Amundsen Gulf is part of the Eastern Beaufort Sea, and it is my understanding they are separate, adjacent water bodies.

Ice

There is still one part of this revised paper that is troubling me, that is the sea ice analysis. The analysis in this version is different than the previous; in this round, they make quantitative comparisons based on a 6 x 6 km grid cell scale, do not appear to define the boundaries of the study area (I may have missed that), and as before, they still base the analysis on only five years of sea ice data. Using these time and space scales is not well relevant to our understanding of bowhead winter ecology and movements, or, to variation in sea ice that is well documented for this region (takes decades to elucidate statistical change/shifts).

Shore to shore to shore fast ice throughout Amundsen Gulf during winter, which is the most common winter condition, could be expected to limit or deter bowheads from entering and overwintering into east Amundsen Gulf. This is inferred from published observations of tagged bowheads 'waiting' at 10/10 ice edges in the Bering Sea and off Cape Bathurst in spring, yet easily navigating cracks etc. in sea ice of 9+ . Therefore, the 6 km x 6 km grid cell scale is not relevant or helpful considering a bowhead whale in winter habitat, while the entirety of the Gulf and ice conditions therein is likely more relevant.

It would be better to say that when there is limited or no fast ice covering the Gulf (e.g., as was the case for the winter of 2018-19 except for 7 days of continuous ice cover (and incidentally in 2017/2018 also), bowheads would have access to eastern Amundsen Gulf in winter. So while winter 2018/2019 sea ice conditions do not appear to have included an extensive or obvious barrier to bowhead overwintering near Ulu, it is an important point that neither did the year before, and neither did about 10 winters over the last four decades.

Specific comments (page numbers relate to the marked up version, not the clean version)

Line 60: Change "Traditional" to "Indigenous"

Line 67 change "to" to "which"

Line 69 change "likely to be" to "could be", or, end sentence at anomaly.

Lines 101, 105, 107, 118 several other instances - is the term manually defined? I'm not exactly sure I know what you mean here. Visual inspection of a computer screen? Measured?

Lines 118-124 : all these additions - could this go in a table? Comes across as a list of data

Line 142, 184: must indicate winters as 2013/2014 etc. etc. don't give just one year for a winter - several instances

Line 147: "always" and "right before break up" - what is the scale of the study area you are referring to? can't possibly be in every 6 km x 6 km grid cell? At that fine scale, how can you say if break up had occurred at a scale relevant to bowheads.

What is the rationale for a grid cell of that size and how is that relevant to bowhead whales.

Line 206: insert number of days not just date

Line 218: ...possible that the same calls could be detected at both of our closest sites.....

Line 219: revise as follows (part struck out is irrelevant here)

.....Cape Bathurst would seem to be the more likely occasional overwintering location, being both a known summertime foraging congregation location for bowhead whales, particularly sub-adults, and a wintertime polynya (27).

Line 233: insert "Gulf" after Amundsen.

Reviewer: 2

Comments to the Author(s)

This paper is much improved compared to the first version. I only have minor comments, pasted below:

Line 16, abstract: I suggest deleting "not completing their annual migration" and sticking with what you have high certainty about, i.e., the fact that bowheads overwintered in what is normally their summering grounds.

Lines 27-28: "which may be related to how dynamic and the magnitude of ..." please improve phrasing

Line 37: "(maximum >200 years)" Such an extreme statement requires a citation.

Line 50: "... and then eastward in early spring as ice leads form," ("in" is missing)

Lines 54-56: somewhat clumsy sentence, suggest at least a comma after "of": "... that we are aware of, that bowheads had ..." Even better would be: "As far as we know, prior to the winter of 2018/19 there is no record or indication, scientific or Traditional Knowledge (...), that bowheads had previously overwintered in the Beaufort Sea."

Line 68: "four locations in the South Amundsen Gulf" See comment below regarding Figure 1.

Line 69: remove "s" on recorders.

Lines 109-113: would be nice to have the percentages of files analyzed for each recorder, much easier to assess in a glance than these numbers.

Lines 114-116: same here, what percentage of the files analyzed at Ulukhaktok are 191 files, etc ... ?

Line 210: isn't "the Amundsen" colloquial?

Line 256: it's always nice when authors acknowledge the reviewers' work

Figure 1: since the place names on the map are duplicated in the legend, which is unnecessary, it seems to me it would be much more helpful to the reader to use one symbol for the 2018-19 recorders and another for the historical recorders.

===PREPARING YOUR MANUSCRIPT===

===PREPARING YOUR REVISION IN SCHOLARONE===

Author's Response to Decision Letter for (RSOS-202268.R0)

See Appendix A.

Decision letter (RSOS-202268.R1)

Dear Dr Insley,

I am pleased to inform you that your manuscript entitled "Bowhead Whales Overwinter in the Amundsen Gulf and Eastern Beaufort Sea" is now accepted for publication in Royal Society Open Science.

on behalf of Dr Denise Greig (Associate Editor) and Kevin Padian (Subject Editor)
openscience@royalsociety.org

Appendix A

Responses to Reviewers: Insley et al Winter Bowheads

Associate Editor Comments to Author (Dr Denise Greig):

Associate Editor: 1

Comments to the Author:

I enjoyed reading this interesting paper and have a few minor questions. I defer to the reviewers experience and knowledge of bowhead whales for a more detailed review and think this will be an interesting addition to the literature once their concerns have been addressed.

I was curious why acoustic recorders were deployed in this region – was bowhead presence suspected?

Acoustic recorders were deployed as part of a long-term study of shipping noise impact in the region. Bowhead whales are known to frequent the area annually during the summer months, as are beluga whales, but are not known to overwinter in the area. They are known to overwinter in the Bering Sea. Our acoustic work aimed to quantify background noise levels, record marine mammal (bowhead and beluga whales and bearded and ringed seals) sounds to determine presence/absence and timing, and record any shipping sounds in order to model and predict shipping noise impact on the marine mammals.

In the discussion on page 10, line 200. Where it says a small number of whales were sighted locally, can you please elaborate here? When were Ulukhahtok whales sighted? Are these sightings backing up your results (i.e. sightings of whales overwintering in their summer feeding area in addition to the sounds)?

This statement is elaborated upon in the Introduction in the paragraph beginning on page 5, line 54, although there is not a lot of detail and not any more detail available. The whales were sighted near the community of Ulukhaktok several times throughout the winter of 2018/19. We learned of these sightings by word of mouth but were not able to speak to the Kitekudlak brothers until July 2019 at which time they confirmed the sightings. Yes, these sightings back up our results. When we heard of the sightings, we were anxious to conduct a detailed search of our acoustic recordings to verify them. However, we were not able to do so until our recorders were recovered during the fall of 2019.

And, finally, I am curious what the significance is of the possible bowhead songs that you recorded.

Bowheads are known to produce rich and variable songs that are likely associated with mating, although this has not been verified, similar to bird song and humpback whale song. The songs are heard during the winter/spring time period and as such for the Bering/Chukchi/Beaufort population of bowhead whales, singing occurs primarily in the Bering and Chukchi Seas. This is the first example that we are aware of bowhead singing behaviour in the eastern Beaufort Sea. Whether this was associated with mating behaviour is possible but unknown. Although we feel it is an interesting finding, and worth following up on in subsequent analyses, alone we did not feel it was sufficiently important to discuss

further in this manuscript.

Reviewer comments to Author:

Reviewer: 1

Comments to the Author(s)

Bowhead Whales Overwinter in the Amundsen Gulf, Eastern Beaufort Sea

Review. January 13, 2021

I was a reviewer of the original submission to Biology Letters, and have carefully reviewed this next version and the authors responses to reviewer comments. In general, I think they have done a good job addressing all of the various comments, and I found that the toning down of the speculation improved the paper.

As written, title implies that Amundsen Gulf is part of the Eastern Beaufort Sea, and it is my understanding they are separate, adjacent water bodies.

The title now reads "...Amundsen Gulf and Eastern Bering Sea."

Ice

There is still one part of this revised paper that is troubling me, that is the sea ice analysis. The analysis in this version is different than the previous; in this round, they make quantitative comparisons based on a 6 x 6 km grid cell scale, do not appear to define the boundaries of the study area (I may have missed that), and as before, they still base the analysis on only five years of sea ice data. Using these time and space scales is not well relevant to our understanding of bowhead winter ecology and movements, or, to variation in sea ice that is well documented for this region (takes decades to elucidate statistical change/shifts).

Shore to shore to shore fast ice throughout Amundsen Gulf during winter, which is the most common winter condition, could be expected to limit or deter bowheads from entering and overwintering into east Amundsen Gulf. This is inferred from published observations of tagged bowheads 'waiting' at 10/10 ice edges in the Bering Sea and off Cape Bathurst in spring, yet easily navigating cracks etc. in sea ice of 9+ . Therefore, the 6 km x 6 km grid cell scale is not relevant or helpful considering a bowhead whale in winter habitat, while the entirety of the Gulf and ice conditions therein is likely more relevant.

It would be better to say that when there is limited or no fast ice covering the Gulf (e.g., as was the case for the winter of 2018-19 except for 7 days of continuous ice cover (and incidentally in 2017/2018 also), bowheads would have access to eastern Amundsen Gulf in winter. So while winter 2018/2019 sea ice conditions do not appear to have included an extensive or obvious barrier to bowhead overwintering near Ulu, it is an important point that neither did the year before, and neither did about 10 winters over the last four decades.

The reviewer makes excellent and relevant points about the importance of a longer and broader perspective on sea ice and we agree completely. Although this is beyond the scope of the current paper, which reports on a specific and important migratory anomaly and the

specific ice conditions associated with it, a far more detailed and broader investigation of ice variation in the region is warranted and one of our next planned pieces of work to carry out. Regarding the specific ice analysis done in our paper, the methods are identical to our previous version, although we have added two more sites on to the comparison, and the 6.25 km grid cell size for the ice data is just the resolution of the satellite data. We took all of these grid cells within a 100 km radius of each of our acoustic recorders and calculated an average. This is outlined in our methods section (lines 138-145).

Specific comments (page numbers relate to the marked up version, not the clean version)

Line 60: Change "Traditional" to "Indigenous"

Done

Line 67 change "to" to "which"

Done

Line 69 change "likely to be" to "could be", or, end sentence at anomaly.

Done

Lines 101, 105, 107, 118 several other instances - is the term manually defined? I'm not exactly sure I know what you mean here. Visual inspection of a computer screen? Measured?

The term has been defined: "Manual" analysis refers to the process of aural and visual signal verification and categorization by a trained analyst, as compared to an automated process using classification algorithms."

Lines 118-124 : all these additions – could this go in a table? Comes across as a list of data This could potentially go in a table but we felt that given the few number of sites, the flow was better maintained by keeping it in sentences. We have edited the sentences. If the editor feels otherwise, we are happy to add a table.

Line 142, 184: must indicate winters as 2013/2014 etc. etc. don't give just one year for a winter – several instances

We have gone through the text to make sure each case is listed correctly as you have noted and made several changes. In the first case noted above (line 142) however, the sentence refers to all winters between 2013 and 2019, or all winters between 2013 and 2018. In these instances we feel it is correct and less burdened as is. If the editor feels otherwise, we are happy to make the change.

Line 147: "always" and "right before break up" - what is the scale of the study area you are referring to? can't possibly be in every 6 km x 6 km grid cell? At that fine scale, how can you say if break up had occurred at a scale relevant to bowheads.

This is an good point and we have revised the sentence to read: "We examined the data between 15 November and 15 April of each winter in order to capture the majority of time between ice formation and ice break-up every year.". As we noted above, the ice analysis is based on an average of the 6 km grid cells within a 100 km radius, so although it was designated to represent localized trends around each recorder, it is a much larger area than 6 km².

What is the rationale for a grid cell of that size and how is that relevant to bowhead whales.

As stated above, the analysis is based on an average of the grids cells within a 100 km radius around the recording sites, not based on the individual 6 km grid cells. Our goal was to characterize the localized ice conditions with respect to each recorder as a way of quantifying the amount of open water available to a bowhead whale in the immediate region and compare this between years. The grid cell size chosen was the finest available from the AMSR-2 satellite array.

Line 206: insert number of days not just date

These are just three examples of specific dates and times where bowheads are detected at two different recording locations, making it unlikely that it is the same individual bowhead being detected at those two sites. We have reworked the text to clarify the point that these are just three examples (lines 196-214).

Line 218: ...possible that the same calls could be detected at both of our closest sites.....

Done

Line 219: revise as follows (part struck out is irrelevant here)

....Cape Bathurst would seem to be the more likely occasional overwintering location, being both a known summertime foraging congregation location for bowhead whales,

particularly sub-adults, and a wintertime polynya (27).

It seems as though the reviewer "struck out" some text that they would like us to delete. However, the formatting has been removed, so we cannot see any text with "strike-through" formatting, so do not know what the reviewer is asking us to do.

Line 233: insert "Gulf" after Amundsen.

Done

Reviewer: 2

Comments to the Author(s)

This paper is much improved compared to the first version. I only have minor comments, pasted below:

Line 16, abstract: I suggest deleting "not completing their annual migration" and sticking with what you have high certainty about, i.e., the fact that bowheads overwintered in what is normally their summering grounds.

Done

Lines 27-28: "which may be related to how dynamic and the magnitude of ..." please improve phrasing

I agree. Sentence changed to read: "...which may be related to differences in environmental variability as well as..."

Line 37: "(maximum >200 years)" Such an extreme statement requires a citation.

A citation is provided at the end of the sentence (#20). If the editor prefers, we could insert the citation after the statement rather than the sentence end.

Line 50: "... and then eastward in early spring as ice leads form," ("in" is missing)
Done (thank you)

Lines 54-56: somewhat clumsy sentence, suggest at least a comma after "of": "... that we are aware of, that bowheads had ..." Even better would be: "As far as we know, prior to the winter of 2018/19 there is no record or indication, scientific or Traditional Knowledge (...), that bowheads had previously overwintered in the Beaufort Sea."

I agree and have inserted your suggested sentence. Thank you.

Line 68: "four locations in the South Amundsen Gulf" See comment below regarding Figure 1.

Rewritten as: "We collected passive acoustic data at four locations in the south Amundsen Gulf during the 2018-2019 winter (Fig. 1) and compared these to several of our previous recordings in the region (see *Comparisons with Other Datasets* below)." Figure 1 has also been edited.

Line 69: remove "s" on recorders.
Done

Lines 109-113: would be nice to have the percentages of files analyzed for each recorder, much easier to assess in a glance than these numbers.
Done

Lines 114-116: same here, what percentage of the files analyzed at Ulukhaktok are 191 files, etc ... ?
Done

Line 210: isn't "the Amundsen" colloquial?
Yes, "Gulf" inserted.

Line 256: it's always nice when authors acknowledge the reviewers' work
I agree! Statement inserted. Thank you for pointing out the omission.

Figure 1: since the place names on the map are duplicated in the legend, which is unnecessary, it seems to me it would be much more helpful to the reader to use one symbol for the 2018-19 recorders and another for the historical recorders.

We have now added the dates that data were available in the map legend. We opt to keep the symbols the same, given that one site had older data, another site had a mix of older data and concurrent data, and then three sites only had 2018-2019 data.